# Effectiveness and cost-effectiveness of Chuna manual therapy for temporomandibular disorder: A randomized clinical trial

Jae-Heung Cho[1], Koh-Woon Kim[1], Hyungsuk Kim[1], Woo-Chul Shin[1], Me-riong Kim[2], Joowon Kim[3], Min-Young Kim[4], Hyun-Woo Cho[5], In-Hyuk Ha[4], Yoon Jae Lee[4]*

1 Department of Korean Rehabilitation Medicine, College of Korean Medicine, Kyung Hee University Korean Medicine Hospital, Kyung Hee University, Seoul, Republic of Korea, 2 Jaseng Hospital of Korean Medicine, Seoul, Republic of Korea, 3 Bucheon Jaseng Hospital of Korean Medicine, Bucheon, Republic of Korea, 4 Jaseng Spine and Joint Research Institute, Jaseng Medical Foundation, Seoul, Republic of Korea, 5 Haeundae Jaseng Hospital of Korean Medicine, Busan, Republic of Korea

* goodsmile8119@gmail.com

## Abstract

The effectiveness and cost-effectiveness of *Chuna* manual therapy (CMT) for temporomandibular joint disorders (TMD) remain unclear. Here, we compared the effectiveness of CMT and usual care for treating myofascial TMD. A 26-week randomized controlled trial was conducted from 2018 to 2019 with 80 patients across five hospitals in Korea who were diagnosed with myofascial TMD and had temporomandibular joint (TMJ) pain lasting more than three months. Patients were randomly assigned in a 1:1 ratio to either the CMT group, which underwent eight sessions of CMT over four weeks, or the usual care (UC) group, which received physical therapy for the same period. Treatment effectiveness was evaluated in terms of pain, function, and quality of life over 26 weeks. For determining cost-effectiveness, quality-adjusted life years (QALY) were analyzed, and the incremental cost-effectiveness ratios from the societal and healthcare system perspectives were calculated. At week 5, the visual analog scale (VAS) scores decreased more in the CMT group than in the control group, although the difference was statistically insignificant. The CMT group showed significant improvement in specific functional and quality of life measures, particularly in the EuroQoL-VAS (-13.21 (95% confidence interval [CI] -20.03 to -6.38) and the Jaw Functional Limitation Scale-Global score of 0.59 (95% CI 0.13 to 1.05), through improvements were not consistent across all indices. The CMT group showed a slightly higher QALY, and the 26-week incremental cost in the CMT group was $338 lower than that of the usual care group. The cost of CMT was $150 higher than that of usual care, and the incremental cost-effectiveness ratio per utility ranged from $4,011 to $17,851. When a "willingness to pay for treatment ($26,375)" threshold was applied, the probability of CMT being cost-effective was 68.1%–98.3%. Despite no significant differences in pain reduction at week 5, CMT was found to be

**Data availability statement:** The data cannot be shared publicly because they contain potentially identifying information. Data are available upon request from Jung-hyun Kwon (jhkwon0302@jaseng.co.kr), the administrative officer of the Institutional Review Board (IRB) at Jaseng Hospital of Korean Medicine. Data will be provide to researchers who meet the criteria for access to confidential data, as determined by the IRB's review and approval, and subsequently made available by the research team.

**Funding:** This research was funded by the Traditional Korean Medicine R&D Program through the Korea Health Industry Development Institute (KHIDI), which is funded by the Ministry of Health & Welfare, Republic of Korea (grant number HB16C0059) (JHC). The funders had no role in study design, data collection and analysis, decision to publish, or preparation of the manuscript.

**Competing interests:** The authors have declared that no competing interests exist.

a cost-effective treatment for TMD, particularly for improving function and quality of life. These findings may serve as a basis for considering the expansion of national health insurance coverage for Chuna therapy in Korea.

**Trial Registration:** Clinical Research Information Service KCT0003192

## Introduction

Temporomandibular joint disorder (TMD) is defined as craniofacial pain involving the temporomandibular joint (TMJ), masticatory muscles, and associated head and neck musculoskeletal structures [1]. Patients with TMD mostly complain of pain, mandibular motion limitations, and TMJ sounds [2]. TMD is also associated with chronic pain conditions such as migraine and fibromyalgia. They also report comorbidity symptoms, such as chronic fatigue, irritable bowel syndrome, and depression, resulting in a substantial impact on their quality of life [3]. The reported prevalence of TMD varies from 5%–12%, but a trend toward an increasing prevalence of TMD was observed in a recent study [4]. Consequently, the interest in the effective treatment for TMD has increased. Conservative treatment is generally suggested as the first-line intervention for TMD and includes education, self-management, physiotherapy, cognitive behavioral therapy, and pharmacotherapy [2]. However, self-reported scores of patients' satisfaction with nonsteroidal anti-inflammatory drugs, occlusal appliances and physical therapy were not significantly different from those of the no treatment group, indicating unsatisfactory treatment effects or improvement [5]. Thus, the interest in complementary and alternative medicine treatment modalities for TMDs, such as acupuncture or manual therapy, has been growing [6], with an increasing amount of evidence from research studies, including systematic review, supporting their effectiveness [7–9].

*Chuna* manual therapy (CMT) is a traditional medicine approach that requires that practitioners use their hands or specific tools to provide effective stimulation to treat physical conditions in the body [10]. CMT is considered a safe and effective treatment option for a wide range of musculoskeletal conditions [11]. In Korea, the national health insurance (NHI) coverage of CMT was implemented in 2019 [12]. The unit cost of CMT varies depending on the technique used, ranging from approximately $19 to $52. However, NHI reimbursement is currently limited to CMT for musculoskeletal diseases classified under the ICD-10 M code and injuries under the S code. Since TMDs are classified under the ICD-10 K07.6 code, CMT for these disorders is not covered by NHI. A cost-effectiveness study of CMT for TMDs is necessary to support the potential expansion of NHI reimbursement to include these disorders. Previous studies have examined the effectiveness of manual therapy for TMD, but a systematic review concluded that the overall evidence was low, with uncertainty based on the risk of bias assessment [13]. Another systematic review assessed the effectiveness of CMT for TMD, but the quality of evidence was low, and most of the randomized controlled trials (RCTs) were conducted in China [7].

Thus far, no clinical studies have reported on the treatment effects and cost-effectiveness of CMT, a Korean traditional manual therapy, for TMD treatment.

Therefore, we conducted this RCT to assess the clinical effectiveness and safety of CMT for TMD and to evaluate its cost-effectiveness through an economic analysis.

## Methods

### Study design

This study was designed as a two-arm, multicenter, assessor-blinded RCT to evaluate the effectiveness and cost-effectiveness of CMT compared to usual care in patients with myofascial TMD. We hypothesized that CMT would result in greater pain reduction, improved jaw function, enhanced health-related quality of life, and favorable cost-effectiveness profile. Additionally, we hypothesized that CMT would lead to faster symptom relief and recovery compared to usual care.

This study was conducted from September 24, 2018, to June 29, 2019, at one university hospital (Kyung Hee University Korean Medicine Hospital, Gangdong) and four spine specialty Korean medicine hospitals (Jaseng Hospital of Korean Medicine in Gangnam, Daejeon, Bucheon, and Haeundae). The protocol of this RCT has been published [14] and was approved by the Institutional Review Board of the respective hospitals (JASENG 2018-06-008, 2018-06-010, 2018-06-011, and 2018-06-012, and KHNMCOH 2018-05-007). Written informed consent was obtained from all participants. This study was conducted according to the Consolidated Standards of Reporting Trials. The study timetable is presented in Table 1.

### Participants

The inclusion criteria were participants aged 19–70 years; numeric rating scale (NRS) score of unilateral or bilateral TMJ pain lasting for more than 3 months; and diagnosis of myofascial TMD Axis I: Group 1 according to the Research Diagnostic Criteria for Temporomandibular Joint Disorders (RDC/TMD). The exclusion criteria were TMJ pain due to trauma, history of TMJ surgery, specific diseases (e.g., rheumatoid arthritis), and use of psychiatric drugs or immunosuppressants. Further details on the inclusion and exclusion criteria for participant eligibility are provided in the protocol publication [14].

### Interventions

Six CMT techniques were selected as interventions in this study, based on the textbook [15] of *Chuna* Manual Medicine. These techniques were applied based on the patient's symptoms and functional impairments. For TMJ-related conditions:

- Sitting TMJ distraction with thumb technique for TMJ displacement and restricted jaw movement

- Sitting lateral pterygoid pushing with index finger technique for lateral pterygoid muscle tension

- Sitting TMJ manipulation with thumb technique for TMJ dislocation

For cervical spine conditions associated with TMD:

- Supine cervical spine distraction technique for cervical tension and stiffness, accompanying TMD

- Supine cervical spine JS distraction manipulation technique for restricted cervical mobility and mild rotational displacement associated with TMD

- Supine cervical spine manipulation technique for cervical displacement observed in TMD patients

Korean medicine doctors (KMDs) selected one or more techniques based on the patient's condition and clinical judgements, recording all applied techniques in electronic medical records and case report forms.

Table 1. Study timetable.

| Time point | Enrollment<br>Week -1 | Allocation<br>Week 0 (Baseline) | Active Treatment post-allocation | | | | Follow-up | | |
|---|---|---|---|---|---|---|---|---|---|
| | | | Week 1 | Week 2 | Week 3 | Week 4 | Week 5 | Week 13 | Week 26 |
| Visit window | | | ±3 | ±3 | ±3 | ±3 | ±3 | ±7 | ±7 |
| Eligibility screening | ○ | | | | | | | | |
| Written Informed consent | ○ | | | | | | | | |
| Vital signs | ○ | | ○ | | | | ○ | | |
| Sociodemographic characteristics, medical history (e.g., TMJ pain, medication history) | ○ | | | | | | | | |
| RDC/TMD test & analysis | ○ | | | | | | | | |
| Randomized allocation | | ○ | | | | | | | |
| TMJ X-RAY | | | ○ | | | | | | |
| Credibility and Expectancy | | | ○ | | | | | | |
| Treatment in Chuna group (experimental group) | | | ← 2 times/week → | | | | | | |
| Treatment in UC group (active control group) | | | ← 2 times/week → | | | | | | |
| Symptoms and change in medicine | | | ○ | ○ | ○ | ○ | ○ | ○ | ○ |
| NRS of TMJ pain/ bothersomeness | ○ | | ← every visit → | | | | ○ | ○ | ○ |
| VAS of TMJ pain | | | ○ | ○ | ○ | ○ | ○ | ○ | ○ |
| K-BDI-2 | | | ○ | | | | ○ | | ○ |
| JFLS | | | ○ | | | | ○ | | |
| TMJ Range of Motion (maximum mouth opening, mandibular excursive movement) | | | ○ | ○ | ○ | ○ | ○ | ○ | ○ |
| PGIC | | | | | | | ○ | ○ | ○ |
| EQ-5D-5L | | | ○ | | | | ○ | ○ | ○ |
| EQ-VAS | | | ○ | | | | ○ | ○ | ○ |
| SF-12 | | | ○ | | | | ○ | ○ | ○ |
| Economic evaluation-Medical costs | | | ○ | | | | ○ | ○ | ○ |
| Economic evaluation-Time costs | | | | ○ | | | | | |
| Economic evaluation-Productivity loss | | | ○ | ○ | ○ | ○ | ○ | ○ | ○ |
| Adverse events | | | ← every visit → | | | | ○ | ○ | ○ |

In the usual care group, commonly used methods of physical therapy[16] for TMJ pain were administered according to the patient's TMD characteristics.

- Interferential current therapy (ICT) was applied for deep muscle tension and myofascial stiffness

- Transcutaneous electrical nerve stimulation (TENS) was prescribed for patients with pain-dominant symptoms

- Thermotherapy and ultrasound therapy were used for general pain relief and muscle relaxation

Both interventions, CMT and usual care, were administered twice weekly over four weeks, with each session consisting of 10 minutes of treatment and 10 minutes of rest, totaling 20 minutes. This study aimed to evaluate the effectiveness of CMT in a real-world setting, adopting a pragmatic approach.

During the treatment period (up to 4 weeks), medications and treatments for TMD from other medical institutions were not allowed, but they were permitted and recorded during the follow-up period. Rescue medications such as acetaminophen was allowed for all participants during the treatment period, with a maximum dosage of 4g per day, and its usage was recorded.

## Outcomes

**Primary outcome.** The primary outcome was the visual analog scale (VAS) score for the TMJ, which was assessed in week 5. In both the Chuna and usual care groups, pain assessments immediately following the interventions may not have been accurate due to intervention-induced sensitivity. Therefore, pain levels were measured one week after completing all assigned interventions, which was set as the primary endpoint. The VAS consisted of a 100-mm line, with one end representing no pain and the other representing the worst imaginable pain. Participants marked the line to indicate their TMD pain intensity over the past week. VAS scores for the TMJ were assessed at baseline, weeks 2, 3, 4, 5, 13 and 26.

**Secondary outcomes.** NRS for TMJ pain & bothersomeness: The Numeric Rating Scale (NRS) was used to objectively assess subjective pain. Patients chose a number from 0 (no pain) to 10 (worst imaginable pain) to represent their discomfort since the last visit. NRS scores for pain and bothersomeness related to TMD were recorded at every visit, including baseline, weeks 5, 13 and 26.

Range of Motion (ROM) of the TMJ: ROM was evaluated at screening, baseline, weeks 2, 3, 4, 5, 13 and 26. Maximum mouth opening without pain, protrusion, and lateral movements were measured using a ROM ruler (Therabite, Sweden). Participants were instructed to open their mouths as widely as possible without pain, and to perform mandibular protrusion and lateral movements. These measurements followed the guidelines set by the international RDC/TMD consortium.

Korean version of Beck's Depression Index-2 (K-BDI-2)[17]: This questionnaire was administered at baseline, week 5 and 26. It consists of 21 questions scored from 0 to 3, with higher scores indicating greater depression severity. A total score of 17 or higher indicated clinical depression, at which point participants were referred to a psychological clinic.

Jaw Functional Limitation Scale (JFLS) [18]: Participants completed this questionnaire at baseline and week 5. The questionnaire assessed jaw function in biting, moving, speaking, and emotional expression, using 20 items rated from 0 (no limitations) to 10 (maximum limitations).

Patient Global Impression of Change (PGIC): The PGIC was assessed week 5, 13 and 26. Participants rated their health improvement on a 7-point Likert scale, ranging from 1 (very much improved) to 7 (very much worse).

Short Form-12 Health Survey (SF-12), version 2 [19]: The SF-12 was completed at baseline, week 5, 13 and 26. This survey contains 12 items assessing health-related quality of life, with higher scores indicating better quality of life.

5-Level EuroQol-5 Dimension (EQ-5D-5L) and EuroQol Visual Analogue Scale (EQ-VAS): Both instruments were administered baseline, week 5, 13 and 26. The EQ-5D-5L evaluates health status across five dimensions (mobility, self-care, usual activities, pain, and anxiety/depression), while the EQ-VAS uses a 100-mm line to rate health, from worst to best possible condition.

Credibility and Expectancy Questionnaire: This 9-point Likert scale was used to measure participants' expectations for the trial's outcome. The Credibility/Expectancy Questionnaire was administered prior to treatment, participants rated how much they expected their symptoms to improve from 1 (not at all) to 9 (very much).

Work Productivity and Activity Impairment (WPAI) Questionnaire[20]: The WPAI was used to assess productivity loss due to TMD. It measured absenteeism, presenteeism, and overall work impairment, as well as impairment in regular activities, for both employed and unemployed participants over the past week. The questionnaire was assessed at baseline, week 2, 3, 4, 13 and 26.

For economic evaluation, healthcare cost data at each visit were investigated, and surveys were conducted to gather information on any concomitant treatments received for TMD during the follow-up period.

## Sample size

The sample size was initially estimated at 34 participants per group based on an effect size of 0.7. Using analysis of covariance (ANCOVA) with baseline data as the covariate and a correlation of 0.3 between baseline and primary endpoint results, the required sample size was adjusted to 62 participants. Accounting for a 30% dropout rate, the final sample size was set at 80 participants, with 40 participants per group.

## Randomization

Block randomization was performed using nQuery Advisor 7.0 software for a clinical study design (Statistical Solutions Ltd, Cork, Ireland). The random allocation results were sealed in opaque envelopes and assigned by enrollment order. They were revealed at participant allocation to ensure concealment throughout the process. The participants were randomly allocated to the CMT or usual care group (1:1 ratio).

## Blinding

Outcome assessors were blinded to group allocation and were not involved in the treatment procedures. They were responsible only for assessing the outcomes, particularly the TMJ range of motion, including maximum mouth opening (MMO; pain-free). Outcome assessors were blinded to group allocation and were not involved in the treatment procedures.

## Statistical analysis

Effectiveness was evaluated by analyzing changes from baseline at each time point for each group. ANCOVA was planned, using baseline values and any covariates showing statistical differences (p-value <0.05) between groups at baseline, with group as the fixed factor. However, no covariates showed significant differences between groups at baseline, so no covariates were included in the final analysis. The analysis was based on the intention-to-treat (ITT) principle, with missing data on costs and utilities handled using multiple imputation. Correlations between baseline covariates and outcomes informed the imputation model [21]. Baseline covariates included treatment allocation, sex, and age. Due to a skewed data distribution, imputations were performed using predictive mean matching and the Markov chain Monte Carlo method, generating 20 imputation sets with the mice package (version 3.6.0) in R version 4.0.1. For sensitivity analysis, a per-protocol (PP) analysis was conducted, including only participants who completed all assigned treatment sessions, with missing data addressed through multiple imputation.

During the study period, AUC analysis and survival analysis were conducted to comprehensively capture the temporal changes in outcomes. The AUC cumulatively reflected the total amount of effectiveness outcomes from randomization to the final follow-up and was calculated on a 1-week basis according to the trapezoidal rule. AUCs were calculated from randomization to the final follow-up, and the cumulative outcome values of the two groups were compared using Student's t-test. Furthermore, the proportion of patients (%) in each group was compared at each timepoint where VAS scores decreased to less than half of the baseline values. Kaplan–Meier curves were employed for survival analysis to compare the time until TMD pain 'recovery' was achieved, defined as pain outcomes decreasing to less than half of the baseline levels post-randomization, and the curves were analyzed using the log-rank test. Hazard ratios between the two groups were compared using the Cox proportional hazards model. A significance level of 0.05 was applied for this study, and all analyses were performed using SAS 9.4 (© SAS Institute, Inc., Cary, NC, USA) and R Studio 1.1.463 (© 2009–2018 RStudio, Inc.).

## Utilities

The quality of life of patients was measured using the EQ-5D-5L, including the EQ-VAS and Short Form 12 Health Survey V2 (SF-12v2) at baseline and five weeks, 13 weeks, and 26 weeks after baseline measurements. For the EQ-5D

questionnaire, the validated Korean version was used, and the scores were converted into a utility index by applying the tariffs presented by Kim et al. [22]. For the SF-12v2 questionnaire, the validated Korean version was used, and the resulting values were converted to the health utility index of SF-6 dimension (SF-6D) using the equation presented by Brazier et al. [23]. The quality-adjusted life years (QALY) were calculated based on the area under the curve (AUC) approach and trapezoid method.

## Perspectives

For economic evaluation and analysis, two perspectives were used. Healthcare costs during the study period were used for conducting healthcare system perspective analysis. For the societal perspective analysis, non-healthcare costs such as time, transportation, and productivity loss costs were considered in addition to the healthcare system costs.

## Unit cost

The cost sources and individual unit costs are detailed in S1 Table. Considered costs included CMT, physical therapy, consultation for doctor visits, and X-rays. Additionally, any healthcare services utilized by patients during the follow-up period were also accounted for. In Korea, CMT is covered by NHI, but not for TMD (a "non-covered service"). Costs can vary by institution. For this analysis, we assumed NHI coverage for CMT for TMD and used the NHI reimbursement rate for calculations. Costs for consultations, physical therapy, and radiography were obtained from the Health Insurance Review & Assessment Service of Korea (2019). No discount rate was applied due to the 26 weeks study period.

## Resource use measurement

In addition to RCT interventions, patients' private healthcare costs related to TMD (e.g., hospital visits and over-the-counter drugs) during follow-up were surveyed using questionnaires at 5 weeks, 13 weeks, and 26 weeks post-randomization. Transportation and time costs were also analyzed and calculated by multiplying total treatment time by sex- and age-stratified income. Productivity loss was measured using WPAI responses [20]. Work impairment was applied to employed patients, and activity impairment to unemployed patients, then multiplied by sex- and age-stratified income [24] to estimate productivity loss costs.

## Uncertainty

For cost-effectiveness analysis, the incremental cost-effectiveness ratio (ICER) was calculated by dividing the cost difference between groups by the utility difference.

To handle uncertainty in the data distribution, 10,000 sample means were extracted using non-parametric bootstrapping. The cost-effectiveness plane was derived by plotting the differences in the extracted costs and QALY, and the probability that an extracted sample was in each quadrant of the plane was calculated. In addition, incremental net benefit and cost-effectiveness acceptability curves were obtained based on Koreans' willingness to pay (WTP) (30,050,000 KRW; 26,375 USD) for their healthcare decisions [25].

The following uncertainties were considered in the sensitivity analysis: 1) Per-protocol (PP) analysis was used to examine uncertainty depending on the clinical study analysis method; 2) regarding analysis from the healthcare system perspective, uncertainty from non-healthcare costs was additionally considered; 3) in calculating the WPAI scores from the societal perspective, the analysis of uncertainty in productivity costs considered costs that only reflected the overall work impairment of those in paid employment; and 4) regarding uncertainty in the limitations of the study period, analyses were performed assuming that the effect of up to 6 months would be maintained for one year.

## Results

Eighty participants were enrolled and included in the ITT analysis (40 each in the CMT and usual care groups), and the number of people who attended the follow-up visit is shown in Fig 1. The number of patients included in the PP analysis, an analysis of the participants who completed all the sessions of the assigned treatment, was 75 (37 in the CMT group and 38 in the usual care group; Fig 1)

## Patient characteristics

No significant differences were found in sex, age, comorbidities, or RDC/TMD group distribution, or baseline scores for VAS, NRS, JFLS, K-BDI, EQ-5D-5L, EQ-VAS, and SF-12 between groups (Table 2). The Credibility/Expectancy Questionnaire (CMT: 7.0±1.5; usual care: 6.5±1.9, p=0.22) was also not statistically significant.

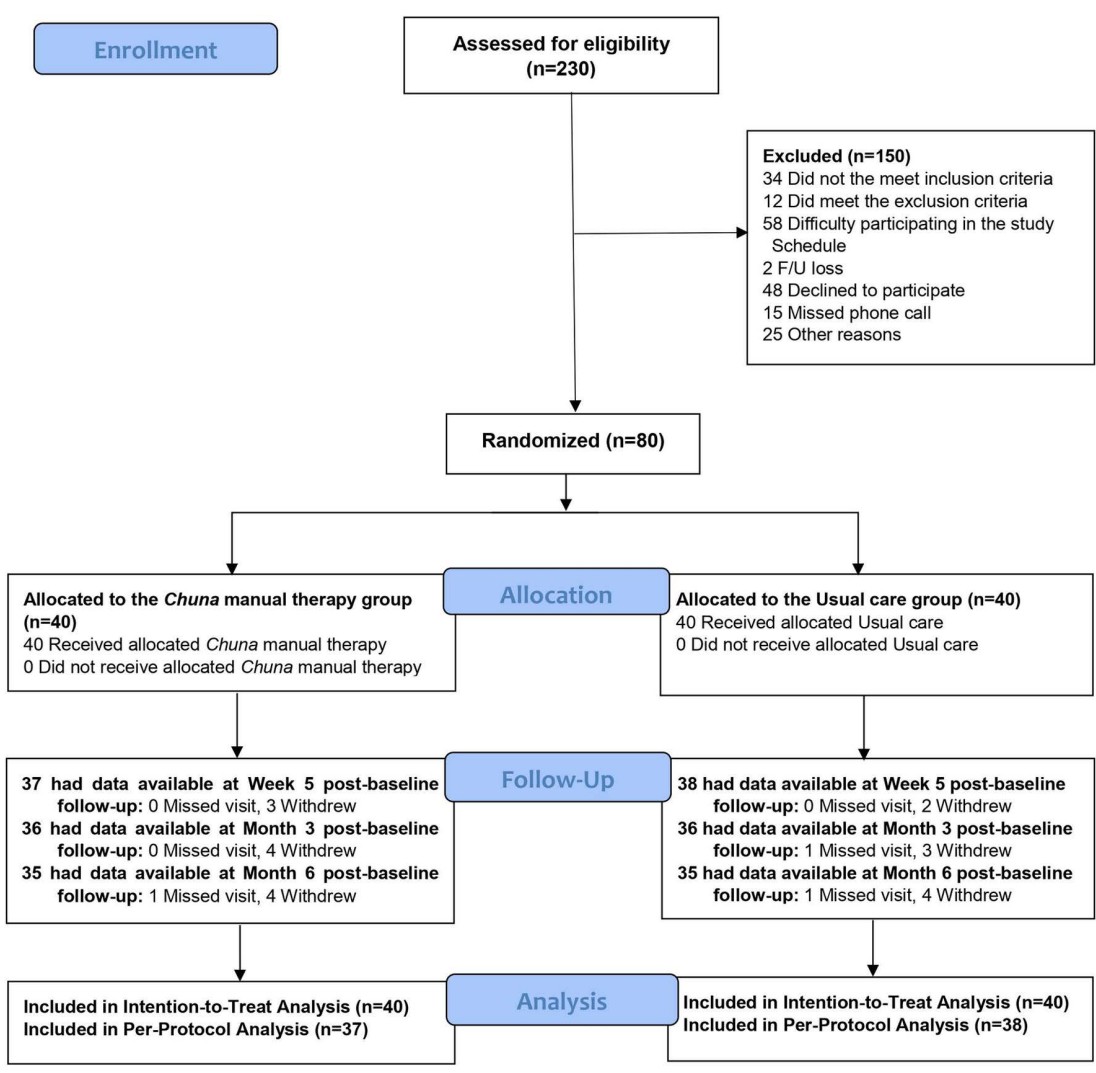

**Fig 1. Study Flowchart.**

**Table 2. Baseline characteristics of participants by randomized group.**

| | | Chuna manual therapy (n = 40) | Usual care (n = 40) | P Value |
|---|---|---|---|---|
| Sex | | | | |
| Female | | 33 (82.5) | 26 (65.0) | .127 |
| Male | | 7 (17.5) | 14 (35.0) | |
| Age, mean (SD), year | | 35.5 (10.2) | 36.0 (10.4) | .829 |
| Height, mean (SD), cm | | 164.8 (7.0) | 166.5 (9.3) | .339 |
| Body weight, mean (SD), kg | | 59.3 (10.4) | 63.2 (13.2) | .147 |
| BMI, mean (SD) | | 21.8 (3.2) | 22.6 (3.1) | .257 |
| Pain duration, mean (SD), mo | | 85.7 (74.3) | 79.8 (60.5) | .696 |
| Comorbidity with other chronic pain | 1 | 22 (55.0) | 19 (47.5) | .655 |
| | 0 | 18 (45.0) | 21 (52.5) | |
| Presence of chronic pain (multiple choice possible) | 1 | 30 (75.0) | 30 (75.0) | 1 |
| | 0 | 10 (25.0) | 10 (25.0) | |
| RDC/TMDc | | | | |
| Group I a | | 19 (47.5) | 22 (55.0) | .655 |
| Group I b | | 21 (52.5) | 18 (45.0) | |
| Group 2 | | 24 (60) | 21 (52.5) | .652 |
| Group 3 | | 29 (72.5) | 20 (50.0) | .063 |
| X-ray (%) | | | | |
| Normal | | 23 (57.5) | 20 (50.0) | .809 |
| Abnormal, NCS | | 4 (10.0) | 5 (12.5) | |
| Abnormal, CS | | 13 (32.5) | 15 (37.5) | |
| VAS, mean (SD), mm | | 52.6 (16.4) | 51.1 (13.7) | .664 |
| NRS for pain, mean (SD), points | | 5.2 (1.2) | 5.2 (1.1) | .924 |
| NRS for bothersomeness, mean (SD), points | | 5.5 (1.7) | 5.9 (1.7) | .317 |
| K-BDI II, mean (SD), points | | 11.3 (7.0) | 11.8 (8.9) | .758 |
| JFLS - Mastication, mean (SD), points | | 4.6 (1.7) | 4.1 (1.8) | .281 |
| JFLS - Mobility, mean (SD), points | | 2.9 (1.4) | 2.8 (1.7) | .802 |
| JFLS - Verbal and emotional, mean (SD), points | | 2.3 (1.7) | 2.1 (2.2) | .651 |
| JFLS - Global, mean (SD), points | | 3.3 (1.4) | 3.0 (1.7) | .506 |
| Maximum mouth opening; range of maximum voluntary mouth opening, mean (SD), mm | | 42.1 (6.3) | 44.6 (8.7) | .153 |
| EQ-5D-5L score, mean (SD), points | | 0.82 (0.08) | 0.82 (0.09) | .669 |
| EQ-VAS, mean (SD), mm | | 58.8 (20.2) | 59.0 (16.6) | .947 |
| SF-12 score, mean (SD), points | | | | |
| PCS | | 45.4 (7.8) | 46.7 (6.1) | .412 |
| MCS | | 47.9 (10.3) | 49.3 (9.1) | .516 |

*RDC/TMD*, Research Diagnostic Criteria for Temporomandibular Joint Disorders; *NCS*, not clinically significant; *CS*, clinically significant; *SD*, standard deviation; *BMI*, body mass index; *VAS*, visual analog scale; *NRS*, numeric rating scale; *JFLS*, jaw functional limitation scale; *EQ-5D-5L*, the EuroQol 5 Dimension 5-level; *SF-12*, the Medical Outcomes Study 12-Item Short-Form Health Survey; *PCS*, Physical Component Summary; *MCS*, Mental Component Summary; *EQ-VAS*, EuroQol-5 dimension visual analog scale; *K-BDI II*, Korean version of Beck's depression index-2

## Treatment

The most frequently used CMT techniques were sitting TMJ distraction with thumb and supine cervical spine JS distraction manipulation technique, administered 7.4±1.6 times per patient. Furthermore, sitting lateral pterygoid pushing with index finger technique and sitting cervical spine distraction techniques were frequently used, with 7.3±1.8 sessions per patient (S2 Table).

Regarding the usual care group, ICT was the most frequently used method (70% of the patients, 7.3±1.8 sessions), followed by deep heat therapy, superficial heat therapy, and TENS.

The list of medications related to TMD treatment (analgesics, such as acetaminophen, used as rescue medication during the intervention period) are presented in S3 Table. During the intervention, medications were used by three participants in the usual care group and one in the CMT group. During follow-up, five participants in the usual care group and one in the CMT group used medications, indicating higher prescription rates in the usual care group.

## Effectiveness and safety

**Effectiveness.** At week 5 post-randomization, the decrease in the VAS scores for TMJ pain was non-significant between groups (Difference in decrease 5.96; 95% CI -1.49 to 13.42).

No significant differences were observed in NRS scores for pain and bothersomeness at week 5, 13, and 26, and similarly, no significant differences were found in VAS scores for TMJ pain.

However, functional outcomes improved significantly in the CMT group: JFLS-Global (0.59; 95% CI 0.13 to 1.05) and JFLS-Verbal and emotional (Difference in decrease 0.86; 95% CI 0.28 to 1.44) at week 5.

Quality of life measures also favored the CMT group: SF-12 PCS (-3.50; 95% CI -5.86 to -1.13) and EQ-VAS (-13.21; 95% CI -20.03 to -6.27) at week 5. By week 13, the SF-12 PCS continued to show improvement (-2.54; 95% CI -4.88 to -0.19) indicating sustained benefits.

The PGIC score showed significant improvement in the CMT group (-0.55; 95% CI -0.91 to -0.18) in week 5 and (-0.47; 95% CI -0.90 to -0.03) in week 13 (Table 3).

Additional outcome comparisons, including ROM, are presented in S4 Table. PP analysis results, consistent with the ITT analysis, are presented in S5 Table. Longitudinal changes in outcomes, including pain(VAS), functional measures (JFLS-Global), and quality of life indicators (SF-12 PCS, EQ-VAS, EQ-5D), are visualized in S1 Figure.

Survival analysis showed significantly faster recovery in the CMT group with a hazard ratio of 1.75 (95% CI 1.01 to 3.03, p = 0.047) based on VAS scores, and the log-rank test also indicated significance (p = 0.028) (S2 Fig). The 26-week AUC comparison indicated significantly faster recovery in terms of function and quality of life in the CMT group, as seen in JFLS-Verbal and emotional, JFLS-Global, K-BDI 2, EQ-VAS, and SF-12 PCS scores (S6 Table).

## Safety

There were three adverse events (AEs) possibly related to the interventions in the CMT group and three in the usual care group, with no statistically significant difference between the two groups. All events were mild in severity, and no serious AEs occurred. The AEs in the CMT group were headache, tinnitus, and oral mucosal swelling, which resolved in 5–6 days. Only oral mucosal swelling resulted in temporary discontinuation of CMT; the headache and tinnitus resolved without treatment. AEs in the usual care group were otalgia, neck pain, and increased TMJ pain, which resolved in 1–13 days. The neck pain and increased TMJ pain resolved after analgesics were administered once a day.

## Economic evaluation

**QALY.** At 26 weeks, QALY, measured using EQ-5D scores, was 0.413 in the CMT group and 0.405 in the usual care group, a difference of 0.0008. Using SF-6D, QALY was 0.369 in the CMT group and 0.352 in the usual care group, a

**Table 3. Primary and secondary outcomes by treatment and time since randomization.**

| Assessment | Categories | Week 5 | Week 13 | Week 26 |
|---|---|---|---|---|
| VAS (Pain) | Chuna manual therapy | 26.14 (20.71 to 31.56) | 27.46 (21.87 to 33.05) | 25.02 (18.63 to 31.42) |
| | Usual care | 32.10 (26.93 to 37.27) | 28.81 (23.19 to 34.43) | 28.05 (21.85 to 34.25) |
| | Difference in decrease | 5.96 (-1.49 to 13.42) | 1.36 (-6.64 to 9.35) | 3.02 (-5.95 to 12.00) |
| | *P*-value | 0.115 | 0.736 | 0.504 |
| NRS (Pain) | Chuna manual therapy | 2.86 (2.30 to 3.43) | 2.99 (2.40 to 3.58) | 2.80 (2.15 to 3.45) |
| | Usual care | 3.39 (2.83 to 3.95) | 3.09 (2.51 to 3.68) | 2.79 (2.14 to 3.45) |
| | Difference in decrease | 0.53 (-0.28 to 1.33) | 0.11 (-0.74 to 0.95) | -0.01 (-0.93 to 0.92) |
| | *P*-value | 0.194 | 0.8 | 0.991 |
| NRS (Bothersomeness) | Chuna manual therapy | 2.97 (2.36 to 3.58) | 3.25 (2.56 to 3.94) | 3.26 (2.50 to 4.01) |
| | Usual care | 3.52 (2.93 to 4.11) | 3.27 (2.60 to 3.95) | 3.21 (2.45 to 3.97) |
| | Difference in decrease | 0.55 (-0.33 to 1.43) | 0.02 (-0.94 to 0.99) | -0.05 (-1.13 to 1.04) |
| | *P*-value | 0.216 | 0.961 | 0.934 |
| EQ-5D-5L | Chuna manual therapy | 0.85 (0.83 to 0.87) | 0.87 (0.84 to 0.89) | 0.87 (0.84 to 0.90) |
| | Usual care | 0.83 (0.81 to 0.85) | 0.84 (0.82 to 0.86) | 0.86 (0.84 to 0.89) |
| | Difference in decrease | -0.02 (-0.05 to 0.01) | -0.03 (-0.06 to 0.00) | 0.00 (-0.05 to 0.04) |
| | *P*-value | 0.253 | 0.084 | 0.806 |
| EQ-VAS | Chuna manual therapy | 72.09 (67.26 to 76.93) | 69.42 (62.79 to 76.05) | 69.30 (62.75 to 75.85) |
| | Usual care | 58.89 (54.15 to 63.62) | 61.31 (54.52 to 68.10) | 64.62 (58.00 to 71.24) |
| | Difference in decrease | -13.21 (-20.03 to -6.38)a | -8.11 (-17.91 to 1.69) | -4.68 (-14.15 to 4.78) |
| | *P*-value | <0.001 | 0.103 | 0.327 |
| PCS (SF-12) | Chuna manual therapy | 49.60 (47.96 to 51.24) | 50.25 (48.61 to 51.90) | 49.58 (47.59 to 51.58) |
| | Usual care | 46.10 (44.48 to 47.72) | 47.71 (46.06 to 49.36) | 49.43 (47.31 to 51.56) |
| | Difference in decrease | -3.50 (-5.86 to -1.13)b | -2.54 (-4.88 to -0.19)c | -0.15 (-3.13 to 2.83) |
| | *P*-value | 0.004 | 0.034 | 0.92 |
| MCS (SF-12) | Chuna manual therapy | 51.49 (49.72 to 53.25) | 52.93 (51.08 to 54.78) | 50.87 (48.13 to 53.61) |
| | Usual care | 51.49 (49.74 to 53.23) | 51.10 (49.25 to 52.94) | 50.19 (47.38 to 53.00) |
| | Difference in decrease | 0.00 (-2.53 to 2.54) | -1.83 (-4.43 to 0.76) | -0.68 (-4.66 to 3.30) |
| | *P*-value | 0.999 | 0.163 | 0.733 |
| PGIC | Chuna manual therapy | 2.49 (2.23 to 2.74) | 2.79 (2.48 to 3.09) | 2.97 (2.64 to 3.31) |
| | Usual care | 3.03 (2.78 to 3.29) | 3.25 (2.95 to 3.56) | 3.27 (2.94 to 3.61) |
| | Difference in decrease | -0.55 (-0.91 to -0.18)b | -0.47 (-0.90 to -0.03) c | -0.30 (-0.78 to 0.18) |
| | *P*-value | 0.004 | 0.037 | 0.213 |
| JFLS - Global | Chuna manual therapy | 2.13 (1.81 to 2.46) | – | – |
| | Usual care | 2.72 (2.40 to 3.04) | – | – |
| | Difference in decrease | 0.59 (0.13 to 1.05) c | – | – |
| | *P*-value | 0.013 | – | – |
| JFLS - Mastication | Chuna manual therapy | 3.02 (2.60 to 3.43) | – | – |
| | Usual care | 3.56 (3.16 to 3.97) | – | – |
| | Difference in decrease | 0.55 (-0.05 to 1.14) | – | – |
| | *P*-value | 0.072 | – | – |
| JFLS - Mobility | Chuna manual therapy | 2.18 (1.75 to 2.61) | – | – |
| | Usual care | 2.53 (2.10 to 2.95) | – | – |
| | Difference in decrease | 0.35 (-0.25 to 0.94) | – | – |
| | *P*-value | 0.248 | – | – |

*(Continued)*

**Table 3.** (Continued)

| Assessment | Categories | Week 5 | Week 13 | Week 26 |
|---|---|---|---|---|
| JFLS - Verbal and emotional | Chuna manual therapy | 1.21 (0.81 to 1.61) | – | – |
| | Usual care | 2.07 (1.68 to 2.46) | – | – |
| | Difference in decrease | 0.86 (0.28 to 1.44) [b] | – | – |
| | *P*-value | 0.004 | – | – |
| K-BDI II | Chuna manual therapy | 7.63 (6.33 to 8.93) | – | 7.23 (5.70 to 8.75) |
| | Usual care | 9.32 (8.06 to 10.59) | – | 9.33 (7.78 to 10.89) |
| | Difference in decrease | 1.69 (-0.20 to 3.58) | – | 2.11 (-0.13 to 4.35) |
| | *P*-value | 0.078 | – | 0.064 |
| Maximum mouth opening without pain | Chuna manual therapy | 43.91 (42.39 to 45.43) | 44.10 (42.49 to 45.71) | 43.85 (42.28 to 45.42) |
| | Usual care | 43.73 (42.19 to 45.26) | 43.74 (42.13 to 45.34) | 43.13 (41.61 to 44.65) |
| | Difference in decrease | -0.18 (-2.41 to 2.04) | -0.36 (-2.72 to 2.00) | -0.72 (-2.95 to 1.51) |
| | *P*-value | 0.871 | 0.76 | 0.523 |

Abbreviations: *CI*, confidence interval; *VAS*, visual analog scale; *NRS*, numeric rating scale; *JFLS*, jaw functional limitation scale; *EQ-5D-5L,* the Euro-Qol 5 Dimension 5-level; *SF-12,* the Medical Outcomes Study 12-Item Short-Form Health Survey; *PCS,* Physical Component Summary; *MCS*, Mental Component Summary; *PGIC*, Patient Global Impression of Change; *EQ-VAS*, EuroQol-5 dimension visual analog scale; *K-BDI II,* Korean version of Beck's depression index-2

[a] *P*<0.001, [b] *P*<0.01, [c] *P*<0.05

difference of 0.016. Using EQ-VAS, QALY was 0.333 in the CMT group and 0.295 in the usual care group, a difference of 0.037 (Table 4).

**Cost.** From the healthcare system perspective, the health cost was $342 in the CMT group and $192 in the usual care group. This difference in the cost mainly increased during the treatment period (5 weeks) in this study. From the societal perspective, including productivity loss, the costs were $5,127 in the CMT group and $5,465 in the usual care group, indicating lower societal costs for the CMT group due to reduced productivity loss over the study period (S7 Table).

**Cost-utility analysis.** From the societal perspective, the CMT group demonstrated EQ-5D, SF-6D, and EQ-VAS QALYs compared with that demonstrated by the control group, and it incurred low costs, making the ICER dominant. From the healthcare system perspective, the ICER calculated using the EQ-5D scores was $17,851/QALY, with a 64.8% 1×WTP probability of cost-effectiveness. The ICER calculated based on the SF-6D scores was $9,113/QALY, with a 94.8% 1×WTP probability of cost-effectiveness. The ICER calculated based on the EQ-VAS scores was $4,011/QALY, which was the lowest of the three, with a 98.3% 1×WTP probability of cost-effectiveness (Table 5 and Fig 2).

In the sensitivity analysis, for scenario 1 based on the PP analysis, the cost difference increased slightly compared to that at baseline. Thus, regarding cost from the healthcare system perspective, the ICER increased slightly, which resulted in a slight decrease in the probability of CMT being cost-effective. In scenario 2, when the cost from the healthcare system perspective was calculated, including non-healthcare costs, the ICER based on the EQ-5D increased slightly, with a slight decrease in the probability of CMT being cost-effective. In scenario 3, in which the cost of productivity loss was calculated by including only those in paid employment, the dominance of CMT was maintained. In scenario 4, with the assumption that the effect of the intervention for up to 6 months of the study would be maintained for one year, the ICER decreased; however, due to uncertainty, the probability of CMT being cost-effective decreased slightly (S8–S10 Tables).

## Discussion

In our study, the comparison of CMT with usual care for TMD revealed no significant difference in pain outcomes. Although the primary outcome of VAS scores at the 5-week point showed no significant difference, a survival analysis

**Table 4. Quality of life years after randomization into the *Chuna* manual therapy and usual care groups.**

| Time | *Chuna* manual therapy | Usual Care | Difference | P Value |
|---|---|---|---|---|
| **EQ-5D** | | | | |
| 5th week | 0.85 (0.83 to 0.87) | 0.83 (0.81 to 0.85) | 0.02 (-0.01 to 0.05) | 0.253 |
| 13th week | 0.87 (0.84 to 0.89) | 0.84 (0.82 to 0.86) | 0.03 (0.00 to 0.06) | 0.084 |
| 26th week | 0.87 (0.84 to 0.90) | 0.86 (0.84 to 0.89) | 0.00 (-0.04 to 0.05) | 0.806 |
| **QALYs (EQ-5D)** | 0.413 (0.405 to 0.421) | 0.405 (0.397 to 0.413) | 0.008 (-0.003 to 0.020) | 0.142 |
| **SF-6D** | | | | |
| 5th week | 0.76 (0.73 to 0.79) | 0.73 (0.70 to 0.75) | 0.04 (0.00 to 0.08) | 0.08 |
| 13th week | 0.78 (0.75 to 0.81) | 0.73 (0.70 to 0.76) | 0.05 (0.00 to 0.09)a | 0.042 |
| 26th week | 0.77 (0.74 to 0.81) | 0.75 (0.71 to 0.78) | 0.02 (-0.02 to 0.07) | 0.318 |
| **QALYs (SF-6D)** | 0.369 (0.359 to 0.379) | 0.352 (0.343 to 0.362) | 0.016 (0.002 to 0.031)a | 0.022 |
| **EQ-VAS** | | | | |
| 5th week | 72.09 (67.26 to 76.93) | 58.89 (54.15 to 63.62) | 13.21 (6.38 to 20.03)b | <0.001 |
| 13th week | 69.42 (62.79 to 76.05) | 61.31 (54.52 to 68.10) | 8.11 (-1.69 to 17.91) | 0.103 |
| 26th week | 69.30 (62.75 to 75.85) | 64.62 (58.00 to 71.24) | 4.68 (-4.78 to 14.15) | 0.327 |
| **QALYs (EQ-VAS)** | 0.333 (0.313 to 0.352) | 0.295 (0.275 to 0.315) | 0.037(0.008 to 0.067)a | 0.012 |

*Abbreviations. **QALY**, quality-adjusted life years; **EQ-5D-5L**, EuroQol 5-Dimension 5-Level; **SF-6D**, Short Form 6-Dimensional Health State; **EQ-VAS**, EuroQol-5 dimension visual analog scale.*

\* QALYs were calculated using the trapezoidal rule. All values are presented as mean and 95% confidence intervals. Differences between the two groups were estimated using the independent t-test.

a *P* < 0.05, b *P* < 0.001

**Table 5. Results of cost-effectiveness analysis of *Chuna* manual therapy compared with usual care<sup>a</sup>.**

| QALY index | Societal perspectives | | | Healthcare system perspectives | | |
|---|---|---|---|---|---|---|
| | EQ-5D-5L | SF-6D | EQ-VAS | EQ-5D-5L | SF-6D | EQ-VAS |
| **ICER ($/QALY)** | Dominant | Dominant | Dominant | 17,851 | 9,113 | 4,011 |
| **Probability of cost-effectiveness by cost-effectiveness plane (%)** | | | | | | |
| Cost-saving + More effective | 65 | 68.7 | 69.1 | — | — | — |
| Cost-increasing + More effective | 28.9 | 30.7 | 30.6 | 93.9 | 99.4 | 99.7 |
| Cost-saving + Less effective | 4.1 | 0.4 | 0 | 6.1 | 0.6 | 0.3 |
| Cost-increasing + Less effective | 2 | 0.2 | 0.3 | — | — | — |
| **Probability of cost-effectiveness at 1xWTP per capita (%)** | 567 (-903–2,089) | 788 (-644–2,325) | 1,332 (-263–2,980) | 70 (-217–374) | 291 (-72–628) | 835 (39–1,594) |
| **Incremental net benefit at 1xWTP per capita ($)** | 78.5 | 85.2 | 95 | 68.4 | 94.8 | 98.3 |

Abbreviations. **QALY**, Quality-adjusted life-years; **ICER**, incremental cost-effectiveness ratio; **EQ-5D-5L**, EuroQol 5-Dimension 5-Level; **SF-6D**, the Short Form 6-Dimensional health state; **WTP**, willingness to pay

<sup>a</sup>For the base case analysis, the QALY was calculated using the EQ-5D-5L. The incremental cost was divided by the incremental QALY to calculate the ICER. After non-parametric bootstrapping, the incremental net benefit and probability of cost-effectiveness were calculated using the 1xWTP threshold ($26,375). The costs from the healthcare system perspective include the costs of formal and informal healthcare involved in chronic neck pain treatment and costs of transportation and time. From the societal perspective, productivity costs from chronic neck pain were included.

defining recovery as a more than 50% reduction in baseline pain demonstrated a hazard ratio of 1.75 (95% CI 1.01 to 3.03, p = 0.047), indicating faster recovery in the CMT group. Chronic TMD is known to limit essential functions such as chewing and speaking, leading to impairments in social activities and quality of life [26,27]. In our study, functional outcomes also improved significantly, particularly in JFLS-Global (0.59, 95% CI 0.13 to 1.05) and JFLS-Verbal and Emotional

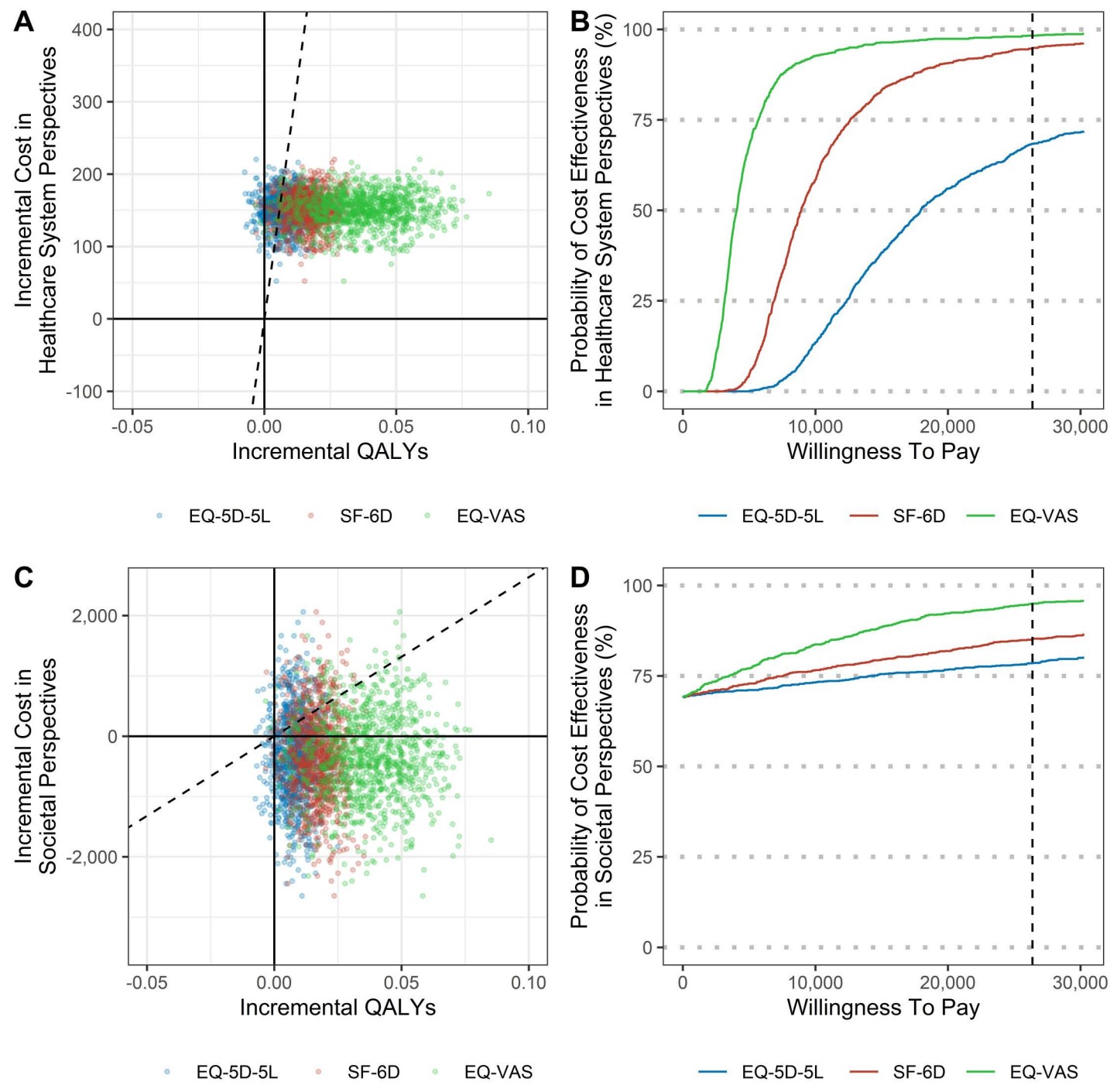

**Fig 2. (A) Cost-effectiveness plane in healthcare system perspectives; (B) Cost-effectiveness acceptability curve in healthcare system perspectives; (C) Cost-effectiveness plane in societal perspectives; and (D) Cost-effectiveness acceptability curve in societal perspectives.**

(0.86, 95% CI 0.28 to 1.44). These improvements in function may have contributed to the observed enhancements in quality of life, as measured by EQ-VAS and SF-12 PCs, in the CMT group. With a comparable number of mild adverse events comparable as the control group, CMT emerges as a safe and effective treatment option for TMD, particularly in improving function and quality of life.

From an economic perspective, CMT was dominant from a societal viewpoint. From the healthcare system perspective, the ICER based on EQ-5D, SF-6D, and EQ-VAS was $17,951/QALY, $9,113/QALY, and $4,011/QALY, respectively, which are below the Korean societal WTP for healthcare. This supports the acceptance of CMT as a viable treatment option in Korea. Currently, NHI covers CMT for musculoskeletal diseases but not for TMD, which falls under a different ICD-10 code. For this study, we assumed NHI coverage for CMT for TMD, specifically using complex CMT involving cervical spine correction, which costs $32, a higher rate compared with the simple CMT cost of $19. Even with this assumption, the cost remained within the Korean WTP, indicating the need to extend NHI coverage to CMT for TMD. If simple *Chuna* therapy were used instead, the ICER would be further reduced, enhancing cost-effectiveness.

Regarding the mechanisms of CMT effectiveness for TMD treatment, manual therapy targets muscles and nerves related to the TMJ, reducing abnormal tension. Additionally, CMT may alleviate TMJ function by improving TMJ and the cervical spine alignment. Cervical spine-related CMT techniques were administered to all the patients in the CMT group. A previous study reported that patients with TMD with neck disabilities had lower satisfaction levels [28]. Thus, improving cervical spine function can enhance TMD treatment satisfaction. Another study found that cervical spine exercises improved pain intensity and pain-free MMO in patients with TMD [29]. Thus, CMT, by aligning the cervical spine and TMJ, may have a more significant impact on functional outcomes and quality of life than on pain outcomes, where no clear significance was found.

A strong contribution of the present study is its attempt to examine the effectiveness of CMT in real-world settings using a pragmatic study design. Another strength is that different sensitivity analyses confirmed the robustness of the analysis in this study. Regarding previous studies on the effectiveness of CMT for TMD, existing systematic reviews have shown that CMT (or Tuina manual therapy) was effective in reducing pain and improving function; however, the quality of evidence in the systematic review was low [7]. Unlike previous systematic reviews, this study did not observe significant differences in pain outcomes, which may be due to differences in the setting and control groups, as the systematic reviews included studies from China. This makes direct comparisons difficult and suggests the need for further research.

The limitations of this study are as follows. According to the RDC/TMD, only participants diagnosed with myofascial TMD, Axis I: Group 1, were included, which limits generalizability of our findings to all TMD types. However, some participants with TMD in RDC/TMD Group 2 (disc displacement) or Group 3 (other joint conditions) were included, suggesting potential effectiveness of CMT for these types as well. Due to the study design, blinding of the physicians (KMDs) and participants was not possible, and outcome assessors were only blinded for specific outcomes, such as the TMJ range of motion, with some outcomes measured through patient-reported questionnaires. In addition, while JFLS scores showed significant differences at week 5, indicating potential functional improvement with CMT, long-term effects could not be analyzed as JFLS scores were only collected at week 5. Moreover, although quality of life and functional improvements were observed, significant effects were found only in specific measures, such as EQ-VAS, SF-12 PCS, and JFLS-Global/Verbal and Emotional scores, rather than across all assessed indices. This suggests that the impact of CMT on overall function and quality of life may not be uniform across all measures. Furthermore, a limitation of this study is that the WTP values used were based on a 2012 survey conducted in Korea, as no recent data were available.

Notwithstanding its limitations, the present study is meaningful in that it is the first RCT to apply CMT in traditional Korean medicine for the treatment of TMD. The findings of this study suggest that, despite no significant differences in pain at week 5, CMT is a cost-effective and effective treatment option for patients with TMD, particularly for those who require improvements in function and quality of life. These results provide a basis for expanding national health insurance coverage of Chuna therapy on TMD in Korea.

## Supporting information

**S1 Fig. Changes in outcomes over time.**
(TIF)

**S2 Fig. Cumulative incidence curves of recovery by group.**
(TIF)

**S1 Table. Cost calculation method, associated data sources, and unit costs.**
(DOCX)

**S2 Table. List of Chuna treatments provided to patients during the intervention period.**
(DOCX)

**S3 Table. List of medication-related to temporomandibular disorders prescribed to patients during the study.**
(DOCX)

**S4 Table. Secondary outcomes by treatment and time since randomization.**
(DOCX)

**S5 Table. Primary and secondary outcomes per protocol analysis with multiple imputations.**
(DOCX)

**S6 Table. Area under the curve of outcomes according to treatment.**
(DOCX)

**S7 Table. Costs per patient after randomization into the chuna manual therapy and usual care groups.**
(DOCX)

**S8 Table. Sensitivity analysis with cost-effectiveness analysis for Chuna manual therapy compared with usual care (EQ-5D-5L).**
(DOCX)

**S9 Table. Sensitivity analysis with cost-effectiveness analysis for Chuna manual therapy compared with usual care (SF-6D).**
(DOCX)

**S10 Table. Sensitivity analysis with cost-effectiveness analysis for Chuna manual therapy compared with usual care (EQ-VAS).**
(DOCX)

**S1 File. Protocol.**
(PDF)

**S2 File. CONSORT checklist.**
(DOCX)

## Author contributions

**Conceptualization:** Jae-Heung Cho, Koh-Woon Kim, In-Hyuk Ha.

**Formal analysis:** Yoon Jae Lee.

**Funding acquisition:** Jae-Heung Cho.

**Investigation:** Koh-Woon Kim, Hyungsuk Kim, Woo-Chul Shin, Me-riong Kim, Joowon Kim, Min-Young Kim, Hyun-Woo Cho.

**Methodology:** Jae-Heung Cho, Koh-Woon Kim, In-Hyuk Ha, Yoon Jae Lee.

**Writing – original draft:** Yoon Jae Lee.

**Writing – review & editing:** In-Hyuk Ha, Yoon Jae Lee.

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
