## [Decision Letter · Decision Letter 0]

13 Aug 2024

PONE-D-24-26791Efficacy and cost-effectiveness of Chuna manual therapy for temporomandibular disorder: A randomized clinical trialPLOS ONE

Dear Dr. Lee,

Thank you for submitting your manuscript to PLOS ONE. After careful consideration, we feel that it has merit but does not fully meet PLOS ONE’s publication criteria as it currently stands. Therefore, we invite you to submit a revised version of the manuscript that addresses the points raised during the review process.

We look forward to receiving your revised manuscript.

Kind regards,

Essam Al-Moraissi

Academic Editor

PLOS ONE

3. Please include a complete copy of PLOS’ questionnaire on inclusivity in global research in your revised manuscript. Our policy for research in this area aims to improve transparency in the reporting of research performed outside of researchers’ own country or community. The policy applies to researchers who have travelled to a different country to conduct research, research with Indigenous populations or their lands, and research on cultural artefacts. The questionnaire can also be requested at the journal’s discretion for any other submissions, even if these conditions are not met.  Please find more information on the policy and a link to download a blank copy of the questionnaire here: https://journals.plos.org/plosone/s/best-practices-in-research-reporting. Please upload a completed version of your questionnaire as Supporting Information when you resubmit your manuscript.

4. To comply with PLOS ONE submissions requirements, in your Methods section, please provide additional information regarding the experiments involving animals and ensure you have included details on methods of anesthesia and/or analgesia, and (3) efforts to alleviate suffering.

6. Thank you for stating the following financial disclosure: 

 [The research work was financially supported by research project (No. PSF/NSLP/KP-UAP (887) funded by Pakistan Science Foundation].  

7. We note that your Data Availability Statement is currently as follows: [All relevant data are within the manuscript and its Supporting Information files.]

Reviewers' comments:

Reviewer's Responses to Questions

**Comments to the Author**

1. Is the manuscript technically sound, and do the data support the conclusions?

Reviewer #1: Yes

Reviewer #2: Yes

2. Has the statistical analysis been performed appropriately and rigorously?

Reviewer #1: Yes

Reviewer #2: Yes

3. Have the authors made all data underlying the findings in their manuscript fully available?

Reviewer #1: Yes

Reviewer #2: Yes

4. Is the manuscript presented in an intelligible fashion and written in standard English?

Reviewer #1: Yes

Reviewer #2: Yes

5. Review Comments to the Author

Reviewer #1: This article presents detailed data and analysis on the efficacy and cost-effectiveness of CMT for TMD. However, compared to the extensive outcome data, there needs to be more sufficient explanation and discussion. Unnecessary data should be excluded, or an explanation and discussion should be added to help the reader understand.

Additionally, I would like to suggest the following comments:

1. The abstract needs to include more information about the trial settings. For example, a sentence like “They underwent eight sessions of CMT and usual care for four weeks” without mentioning the group assignment confuses how the interventions and control were designed. We recommend adding information about the methods to the abstract and describing only the key points in the results section.

2. In the introduction, it is not appropriate to express the content of reference number 5 as “conservative treatments,” and the intervention applied in that study should be specifically expressed. In addition, the evidence for other treatments should not be determined based on only a few research results; more comprehensive evidence should be presented.

3. The method of CMT and physical therapy is not specifically presented. There should be a specific description that allows the reader to reproduce the procedure. In addition, since the treatment applied to each patient is different, a specific explanation of the criteria for why the treatment is applied differently to each patient is necessary.

4. The evidence for expressing the physical therapy in the control group as “commonly used methods” should also be described.

5. The trial timeline needs to be clearly recognized. Please present the timeline in a figure or table and unify the expression of each time point in the manuscript.

6. The manuscript's structure does not comply with CONSORT. Please appropriately reorganize the contents of 'Methods' and 'Results' using the headings and subheadings. In particular, please clearly distinguish between the contents on efficacy and cost-effectiveness.

7. Moving the baseline demographics in Table S2 into the main text seems appropriate.

8. To increase visibility, I suggest organizing the results in Table 1 into separate tables by outcome measures.

9. The primary outcome measure for the research hypothesis and the other secondary outcome measures are not distinguished throughout the study. The results section should be described in separate paragraphs by outcome measures. In addition, in terms of conclusion, the results of the primary outcome measure should be presented first.

Reviewer #2: The authors demonstrated the efficacy and cost-effectiveness of Chuna manual therapy compared to usual therapy in patients with temporomandibular disorder. This study has provided strengthened evidence for the clinical effectiveness and cost-effectiveness of Chuna manual therapy for these patients through randomized controlled trials, which have had limited evidence so far. However, some clarifications are needed to ensure the robustness of this evidence.

Major comments

First, there is a lack of explanation on how various pieces of information, such as clinical outcomes, utilities, and costs, are connected to the conclusions. For example, in this study, there was no significant difference in pain outcomes between Chuna manual therapy and usual care, yet the discussion highlights the reduction of pain as a strength of this study. A more logical explanation of this aspect should be provided. Additionally, the conclusion in the abstract states that Chuna manual therapy is a safer option compared to usual care, but it does not mention whether there is any statistical significance between the two groups.

Secondly, while the paper includes a lot of information on clinical outcomes, utilities, and costs, it lacks detailed explanations for each piece of information, making it difficult to obtain sufficient and specific details. For example, it is unclear what treatments are included in Chuna manual therapy or usual care, and whether patients received more than one treatment. Although this can be found in the supplemental table, it is crucial for understanding the intervention and should be mentioned in the methods section. Additionally, while the QOL outcomes are presented, it is necessary to explain the score ranges of each tool and whether a higher score indicates a higher or lower QOL to properly understand the results.

Thirdly, it is recommended to reorganize the sequence of the paper’s flow. The main points that the reader needs to know should be presented first, followed by the supporting evidence, to help the reader better understand the content.

Specific comments for each section

Abstract

1) Lines 35-36: Recommend including the WTP value.

Introduction

1) Lines 58-61: This part should be included in the methods section to explain the unit costs of CMT. Additionally, it is necessary to provide more details on why a cost-effectiveness study is needed (e.g., for reimbursement from NHI).

Methods

1) Lines 90-91: Which outcome was the assessor not involved in? Please clarify this. Additionally, since most QOL tools are measured by self-reported questionnaires, how is assessor blinding meaningful in this context?

2) Lines 94-96: What techniques of CMT were included in this RCT? Please provide a potential list of CMT techniques here, or the exact list if it has already been selected.

3) Lines 97-99: Please clarify whether a patient can receive only one treatment from the usual care list or more than one treatment from the list.

4) Line 101: Please mention the medications that were possibly administered to both groups.

5) Line 103: Provide the rationale for why the primary outcome was assessed at 5 weeks rather than at a longer follow-up.

6) Lines 103-108: Provide detailed information about the outcome measures (e.g., range of scores, questionnaire composition, what higher scores mean) to understand the meaning of the scores.

7) Lines 119-124: Which items were included in the costs (e.g., doctor’s visits)? Please add the list of items that were included in the cost estimate.

8) Lines 132-136: Recommend moving this section before the 'Unit Cost' section so that readers can understand the perspectives first and then learn how to calculate costs according to the two perspectives.

9) Lines 142-145: Please revise the sentence to correct the grammar. Also, clarify which covariates were added and the meaning of significance (e.g., significance level of 0.05 based on the p-value).

10) Lines 157-158: The results of the WTP were estimated from a publication in 2012, which was 12 years ago. Are they still applicable in 2024?

Results

1) Lines 174-175: What do the scores of CMT 7.0 and usual care 6.5 refer to? Present the p-value indicating that there is no significant difference between the two groups.

2) Lines 183-187: The method does not mention that medication can be used for both the intervention and comparator. Please add this information.

3) Lines 190-197: Since the method does not explain the scores for each outcome measure, it is difficult to understand the meaning of the direction and magnitude of each score. I recommend adding this information to both the methods section and the footnote of the table. Additionally, it is necessary to provide more explanation in the discussion to derive conclusions about the implications of the scores for various clinical outcomes.

4) Lines 216-219: Please add the meaning of 'faster recovery' and specify what the survival analysis pertains to.

5) Lines 248-251: Please use the correct unit for ICER ($/QALY) throughout the manuscript.

Discussion

1) Lines 307-309: Same comments as in Methods 1). The outcome measures include self-reported questionnaires. In this case, can we say that assessor-blinding is useful?

2) Lines 275-280: Pain showed non-significant results, but function improved, and it is understood that the quality of life (QOL) improved as a result. Please provide a more detailed explanation of this by discussing the meaning of the QOL measures for each dimension.

3) Lines 291-292: In the results of this study, significant results for pain were not observed, yet there is a discrepancy with the reported pain improvement in the discussion. To maintain consistency between the study results and the discussion, it is necessary to thoroughly address this discrepancy.

4) Lines 300-302: Similar to the previous question, please explain the discrepancies regarding reducing pain in relation to the results of this study.

5) Lines 314-315: What is the meaning of a conservative treatment option? Also, Draw meaningful conclusions based on all the outcomes included in the results.

6. PLOS authors have the option to publish the peer review history of their article (what does this mean? ). If published, this will include your full peer review and any attached files.

**Do you want your identity to be public for this peer review?** For information about this choice, including consent withdrawal, please see our Privacy Policy .

Reviewer #1: No

Reviewer #2: No

---

## [Author Response · Author response to Decision Letter 1]

1 Oct 2024

-> I have checked your style requirements.

-> I have revised it.

[This research was funded by the Traditional Korean Medicine R&D Program through the Korea Health Industry Development Institute (KHIDI), which is funded by the Ministry of Health & Welfare, Republic of Korea (grant number HB16C0059).].

-> This research was funded by the Traditional Korean Medicine R&D Program through the Korea Health Industry Development Institute (KHIDI), which is funded by the Ministry of Health & Welfare, Republic of Korea (grant number HB16C0059) (JHC). The funders had no role in study design, data collection and analysis, decision to publish, or preparation of the manuscript.

-> The data cannot be shared publicly because they contain potentially identifying information. Data are available upon request from Jung-hyun Kwon (jhkwon0302@jaseng.co.kr), the administrative officer of the Institutional Review Board (IRB) at Jaseng Hospital of Korean Medicine. Data will be provide to researchers who meet the criteria for access to confidential data, as determined by the IRB’s review and approval, and subsequently made available by the research team.

5. For studies involving third-party data, we encourage authors to share any data specific to their analyses that they can legally distribute. PLOS recognizes, however, that authors may be using third-party data they do not have the rights to share. When third-party data cannot be publicly shared, authors must provide all information necessary for interested researchers to apply to gain access to the data. (https://journals.plos.org/plosone/s/data-availability#loc-acceptable-data-access-restrictions)

4) All necessary contact information others would need to apply to gain access to the data.

Reviewer #1: This article presents detailed data and analysis on the efficacy and cost-effectiveness of CMT for TMD. However, compared to the extensive outcome data, there needs to be more sufficient explanation and discussion. Unnecessary data should be excluded, or an explanation and discussion should be added to help the reader understand.

Additionally, I would like to suggest the following comments:

1. The abstract needs to include more information about the trial settings. For example, a sentence like “They underwent eight sessions of CMT and usual care for four weeks” without mentioning the group assignment confuses how the interventions and control were designed. We recommend adding information about the methods to the abstract and describing only the key points in the results section.

-> Thank you for your valuable feedback. We agree that the description of the trial settings needed clarification. We have revised the abstract to specify the random assignment of patients to the intervention and control groups and how the interventions were designed.

Patients were randomly assigned to either the CMT group, which underwent eight sessions of CMT over four weeks, or the usual care (UC) group, which received physical therapy for the same period, in a 1:1 ratio.

2. In the introduction, it is not appropriate to express the content of reference number 5 as “conservative treatments,” and the intervention applied in that study should be specifically expressed. In addition, the evidence for other treatments should not be determined based on only a few research results; more comprehensive evidence should be presented.

-> I appreciate your valuable comments. I have revised the manuscript as follows and added the references.

However, self-reported scores of patients’ satisfaction with nonsteroidal anti-inflammatory drugs, occlusal appliances and physical therapy were not significantly different from those of the no treatment group, indicating unsatisfactory treatment effects or improvement [5]. Thus, the interest in complementary and alternative medicine treatment modalities for TMDs, such as acupuncture or manual therapy, has been growing [6], with an increasing amount of evidence from research studies, including systematic review, supporting their effectiveness [7-9].

3. The method of CMT and physical therapy is not specifically presented. There should be a specific description that allows the reader to reproduce the procedure. In addition, since the treatment applied to each patient is different, a specific explanation of the criteria for why the treatment is applied differently to each patient is necessary.

Thank you for comments. I have revised it as follows.

These included three techniques for the TMJ (Sitting TMJ distraction with thumb technique, Sitting lateral pterygoid pushing with index finger technique, and Sitting TMJ manipulation with thumb technique) and three techniques for the cervical spine (Supine cervical spine distraction technique, Supine cervical spine JS distraction manipulation technique and Supine cervical spine manipulation technique). Korean medicine doctors (KMDs) administered one or more of the selected CMT techniques based on the patient’s condition and their clinical judgments.

This study aimed to evaluated the effectiveness of CMT in a real-world setting, adopting a pragmatic approach. The choice of CMT techniques or types of physical therapy was determined by the physician based on the patients’ clinical condition.

4. The evidence for expressing the physical therapy in the control group as “commonly used methods” should also be described.

-> I have added the reference.

Abouelhuda AM, Khalifa AK, Kim Y-K, Hegazy SA. Non-invasive different modalities of treatment for temporomandibular disorders: review of literature. Journal of the Korean Association of Oral and Maxillofacial Surgeons. 2018;44(2):43.

5. The trial timeline needs to be clearly recognized. Please present the timeline in a figure or table and unify the expression of each time point in the manuscript.

-> I appreciated your valuable comments. I have added the table of timetable, which has been included as Table 1.

5. The manuscript's structure does not comply with CONSORT. Please appropriately reorganize the contents of 'Methods' and 'Results' using the headings and subheadings. In particular, please clearly distinguish between the contents on efficacy and cost-effectiveness.

-> I reorganized the subheadings in the Methods section to comply with CONSORT and I have distinguished between the contents on efficacy and cost-effectiveness.

6. Moving the baseline demographics in Table S2 into the main text seems appropriate.

-> Table S2 has been moved into the manuscript as Table 2. Thank you for your suggestion.

8. To increase visibility, I suggest organizing the results in Table 1 into separate tables by outcome measures.

-> I have revised the format of Table 1 (now changed to Table 3). Please review the updated Table 3.

9. The primary outcome measure for the research hypothesis and the other secondary outcome measures are not distinguished throughout the study. The results section should be described in separate paragraphs by outcome measures. In addition, in terms of conclusion, the results of the primary outcome measure should be presented first.

-> Thank you for your suggestion. The results section has been revised according to your recommendation.

At week 5 post-randomization, the decrease in the VAS scores for TMJ pain was non-significant between groups (Difference in decrease 5.96; 95% CI -1.49 to 13.42).

No significant differences were observed in NRS scores for pain and bothersomeness at week 5, 13, and 26, and similarly, no significant differences were found in VAS scores for TMJ pain.

However, functional outcomes improved significantly in the CMT group: JFLS-Global (0.59; 95% CI 0.13 to 1.05) and JFLS-Verbal and emotional (Difference in decrease 0.86; 95% CI 0.28 to 1.44) at week 5.

Quality of life measures also favored the CMT group: SF-12 PCS (-3.50; 95% CI -5.86 to -1.13) and EQ-VAS (-13.21; 95% CI -20.03 to -6.27) at week 5. By week 13, the SF-12 PCS continued to show improvement (-2.54; 95% CI -4.88 to -0.19) indicating sustained benefits.

The PGIC score showed significant improvement in the CMT group (-0.55; 95% CI -0.91 to -0.18) in week 5 and (-0.47; 95% CI -0.90 to -0.03) in week 13 (Table 3).

Additionally, the conclusion has been changed in both the abstract and in discussion.

The findings of this study suggest that, despite no significant differences in pain at week 5, CMT is a cost-effective and effective treatment option for patients with TMD, particularly for those who require improvements in function and quality of life.

Reviewer #2: The authors demonstrated the efficacy and cost-effectiveness of Chuna manual therapy compared to usual therapy in patients with temporomandibular disorder. This study has provided strengthened evidence for the clinical effectiveness and cost-effectiveness of Chuna manual therapy for these patients through randomized controlled trials, which have had limited evidence so far. However, some clarifications are needed to ensure the robustness of this evidence.

Major comments

First, there is a lack of explanation on how various pieces of information, such as clinical outcomes, utilities, and costs, are connected to the conclusions. For example, in this study, there was no significant difference in pain outcomes between Chuna manual therapy and usual care, yet the discussion highlights the reduction of pain as a strength of this study. A more logical explanation of this aspect should be provided. Additionally, the conclusion in the abstract states that Chuna manual therapy is a safer option compared to usual care, but it does not mention whether there is any statistical significance between the two groups.

I apologize for the confusion caused by the mixed terminology in the discussion of fast recovery through survival analysis. The incorrect part mentioned in the discussion has been corrected with additional explanation. Furthermore, I have added the information regarding the lack of significant differences in AE occurrences under the safety subheading."

Secondly, while the paper includes a lot of information on clinical outcomes, utilities, and costs, it lacks detailed explanations for each piece of information, making it difficult to obtain sufficient and specific details. For example, it is unclear what treatments are included in Chuna manual therapy or usual care, and whether patients received more than one treatment. Although this can be found in the supplemental table, it is crucial for understanding the intervention and should be mentioned in the methods section. Additionally, while the QOL outcomes are presented, it is necessary to explain the score ranges of each tool and whether a higher score indicates a higher or lower QOL to properly understand the results.

I fully agree your comments. As per your suggestions, I have added detailed information about the intervention in the methods section.

Thirdly, it is recommended to reorganize the sequence of the paper’s flow. The main points that the reader needs to know should be presented first, followed by the supporting evidence, to help the reader better understand the content.

I have reorganized the method and the results section. I greatly appreciate your valuable comments.

Specific comments for each section

Abstract

1) Lines 35-36: Recommend including the WTP value.

-> I have added the WPT value ($26,375) in the abstract as per your suggestion.

Introduction

1) Lines 58-61: This part should be included in the methods section to explain the unit costs of CMT. Additionally, it is necessary to provide more details on why a cost-effectiveness study is needed (e.g., for reimbursement from NHI).

-> Thank you for your comment. We have revised the manuscript to move the explanation of the NHI coverage of Chuna Manual Therapy (CMT) to the Methods section, including details about the unit costs of CMT ($21–$54 depending on the technique). Additionally, we have expanded the rationale for the cost-effectiveness study, emphasizing that it is essential to support potential reimbursement for temporomandibular disorders (TMDs) under the NHI system.

In Korea, the national health insurance (NHI) coverage of CMT was implemented in 2019 [12]. The unit cost of CMT varies depending on the technique used, ranging from approximately $19 to $52. However, NHI reimbursement is currently limited to CMT for musculoskeletal diseases classified under the ICD-10 M code and injuries under the S code. Since TMDs are classified under the ICD-10 K07.6 code, CMT for these disorders is not covered by NHI. A cost-effectiveness study of CMT for TMDs is necessary to support the potential expansion of NHI reimbursement to include these disorders.

Methods

1) Lines 90-91: Which outcome was the assessor not involved in? Please clarify this. Additionally, since most QOL tools are measured by self-reported questionnaires, how is assessor blinding meaningful in this context?

-> Based on your comments, the following revisions have been made.

Outcome assessors were blinded to group allocation and were not involved

---

## [Decision Letter · Decision Letter 1]

22 Jan 2025

PONE-D-24-26791R1Efficacy and cost-effectiveness of Chuna manual therapy for temporomandibular disorder: A randomized clinical trialPLOS ONE

Dear Dr. Lee,

Thank you for submitting your manuscript to PLOS ONE. After careful consideration, we feel that it has merit but does not fully meet PLOS ONE’s publication criteria as it currently stands. Therefore, we invite you to submit a revised version of the manuscript that addresses the points raised during the review process.

** both Reviewers have been recommended  further minor revisions. Kindly address these minor revision before final consideration for your paper**

We look forward to receiving your revised manuscript.

Kind regards,

Essam Al-Moraissi

Academic Editor

PLOS ONE

Journal Requirements:

Reviewers' comments:

Reviewer's Responses to Questions

**Comments to the Author**

1. If the authors have adequately addressed your comments raised in a previous round of review and you feel that this manuscript is now acceptable for publication, you may indicate that here to bypass the “Comments to the Author” section, enter your conflict of interest statement in the “Confidential to Editor” section, and submit your "Accept" recommendation.

Reviewer #1: All comments have been addressed

Reviewer #3: (No Response)

2. Is the manuscript technically sound, and do the data support the conclusions?

Reviewer #1: Yes

Reviewer #3: Yes

3. Has the statistical analysis been performed appropriately and rigorously?

Reviewer #1: Yes

Reviewer #3: Yes

4. Have the authors made all data underlying the findings in their manuscript fully available?

Reviewer #1: Yes

Reviewer #3: Yes

5. Is the manuscript presented in an intelligible fashion and written in standard English?

Reviewer #1: Yes

Reviewer #3: Yes

6. Review Comments to the Author

Reviewer #1: The manuscript has been appropriately revised overall, but some parts require supplementation. Here are some suggestions:

1. Revise the “Study timetable” table that appears twice.

2. Please indicate the exact book “textbook of Chuna Manual Medicine” as a reference.

3. “Efficacy” and “effectiveness” are used interchangeably throughout the manuscript. Considering the pragmatic study design, please consider whether it would be more appropriate to use “effectiveness” rather than “Efficacy.”

4. The criteria for treatment selection still need to be clarified. Six CMT techniques are presented, but the symptoms or criteria for each technique applied should be described in the methods or discussion part. The same explanation is required for physical therapy in the control group, and in particular, TENS and ICT were applied to different patients, so criteria for distinguishing them should be provided.

5. Please indicate the statistically significant results in Table 3.

6. There was a significant improvement in some of the functional indices (JFLS, ROM) and quality of life indices (EQ-5D-5L, EQ-VAS, SF-12). However, expressing that the overall function and quality of life improved is not appropriate. It is necessary to describe exactly which functions and quality of life improved in a limited way.

7. The “7.3±0” value in the S2 Table seems to be an error.

Reviewer #3: This review concentrates primarily on the statistical analysis. In general, the analysis seems sound. However, the statistical methods section should be rewritten slightly to clarify the various components of the statistical analysis.

Was the AUC used as a way of time-averaging the endpoint(s)? Please motivate the use of the AUC in this setting.

More generally, note that each study assessment gives rise to one or more endpoints, which are used to assess the hypotheses of the study. With this in mind, please indicate the statistical methods section which endpoints are assessed via which statistical methods, as well as any transformations that are applied before statistical analysis (e.g., the AUC). This should help clarify the flow from data to results to conclusions.

To further elaborate, generally, the structure should be rather parallel, with:

Hypotheses -> Assessments -> Endpoints -> Statistical methods -> Results -> Conclusions

It would be useful to have figures that display longitudinal data --- these could be included just as supplementary figures.

7. PLOS authors have the option to publish the peer review history of their article (what does this mean? ). If published, this will include your full peer review and any attached files.

**Do you want your identity to be public for this peer review?** For information about this choice, including consent withdrawal, please see our Privacy Policy .

Reviewer #1: No

Reviewer #3: No

---

## [Author Response · Author response to Decision Letter 2]

3 Mar 2025

Journal Requirements:

> We have thoroughly checked the reference list and confirmed that there are no retracted references cited in the manuscript. However, in response to the reviewer's comments, we have added one additional reference, which resulted in updates to the reference numbering and citations throughout the manuscript.

Reviewers' comments:

Reviewer's Responses to Questions

Comments to the Author

1. If the authors have adequately addressed your comments raised in a previous round of review and you feel that this manuscript is now acceptable for publication, you may indicate that here to bypass the “Comments to the Author” section, enter your conflict of interest statement in the “Confidential to Editor” section, and submit your "Accept" recommendation.

Reviewer #1: All comments have been addressed

Reviewer #3: (No Response)

2. Is the manuscript technically sound, and do the data support the conclusions?

Reviewer #1: Yes

Reviewer #3: Yes

3. Has the statistical analysis been performed appropriately and rigorously?

Reviewer #1: Yes

Reviewer #3: Yes

4. Have the authors made all data underlying the findings in their manuscript fully available?

Reviewer #1: Yes

Reviewer #3: Yes

5. Is the manuscript presented in an intelligible fashion and written in standard English?

Reviewer #1: Yes

Reviewer #3: Yes

6. Review Comments to the Author

Reviewer #1: The manuscript has been appropriately revised overall, but some parts require supplementation. Here are some suggestions:

1. Revise the “Study timetable” table that appears twice.

Thank you for your careful review and valuable feedback. We apologize for the oversight regarding the duplication of the “Study timetable” table. We have now corrected this by removing the redundant table. We appreciate your attention to detail and your constructive suggestions.

2. Please indicate the exact book “textbook of Chuna Manual Medicine” as a reference.

Thank you for your valuable feedback. We have now explicitly cited the exact book, Textbook of Chuna Manual Medicine, as a reference in the manuscript. We appreciate your careful review and constructive suggestions.

3. “Efficacy” and “effectiveness” are used interchangeably throughout the manuscript. Considering the pragmatic study design, please consider whether it would be more appropriate to use “effectiveness” rather than “Efficacy.”

Thank you for your insightful comment. We agree with your suggestion and have revised the manuscript to consistently use “effectiveness” instead of “efficacy,” considering the pragmatic study design. We appreciate your careful review and constructive feedback.

4. The criteria for treatment selection still need to be clarified. Six CMT techniques are presented, but the symptoms or criteria for each technique applied should be described in the methods or discussion part. The same explanation is required for physical therapy in the control group, and in particular, TENS and ICT were applied to different patients, so criteria for distinguishing them should be provided.

Thank you for your valuable comment. We have revised the Methods section to clarify the criteria for treatment selection. Specifically, we have detailed the six CMT techniques and their corresponding indications based on the Textbook of Chuna Manual Medicine. These techniques were applied according to the patient's symptoms and functional impairments, with Korean Medicine Doctors (KMDs) selecting and documenting the applied techniques in electronic medical records and case report forms.

Additionally, we have provided further clarification on the usual care group, specifying the selection criteria for physical therapy modalities, including Interferential Current Therapy (ICT) and Transcutaneous Electrical Nerve Stimulation (TENS). ICT was applied for deep muscle tension and myofascial stiffness, while TENS was prescribed for patients with pain-dominant symptoms. Thermotherapy and ultrasound therapy were used for general pain relief and muscle relaxation.

For more details, please refer to the revised Methods section. We appreciate your constructive feedback, which has helped improve the clarity and precision of our manuscript.

5. Please indicate the statistically significant results in Table 3.

Thank you for your suggestion. We have now indicated the statistically significant results in Table 3 using the following notation: a P < 0.001, b P < 0.01, c P < 0.05.

6. There was a significant improvement in some of the functional indices (JFLS, ROM) and quality of life indices (EQ-5D-5L, EQ-VAS, SF-12). However, expressing that the overall function and quality of life improved is not appropriate. It is necessary to describe exactly which functions and quality of life improved in a limited way.

Thank you for your valuable feedback. In response to your comment, we have revised the manuscript to specify the functional and quality of life measures that showed significant improvements rather than making a generalized statement about overall function and quality of life.

• In the abstract, we now state:

"The CMT group showed significant improvement in specific functional and quality of life measures, particularly in the EuroQoL-VAS (-13.21 (95% confidence interval [CI] -20.03 to -6.38) and the Jaw Functional Limitation Scale-Global score of 0.59 (95% CI 0.13 to 1.05), though improvements were not consistent across all indices."

• In the Discussion section, we have clarified:

"Functional outcomes also improved significantly, particularly in JFLS-Global (0.59, 95% CI 0.13 to 1.05) and JFLS-Verbal and Emotional (0.86, 95% CI 0.28 to 1.44). These improvements in function may have contributed to the observed enhancements in quality of life, as measured by EQ-VAS and SF-12 PCS, in the CMT group."

• Additionally, in the Limitations section, we have explicitly noted:

"Moreover, although quality of life and functional improvements were observed, significant effects were found only in specific measures, such as EQ-VAS, SF-12 PCS, and JFLS-Global/Verbal and Emotional scores, rather than across all assessed indices. This suggests that the impact of CMT on overall function and quality of life may not be uniform across all measures."

These revisions ensure that our conclusions accurately reflect the findings without overstating the results. We appreciate your constructive comments, which have helped improve the precision and clarity of our manuscript.

7. The “7.3±0” value in the S2 Table seems to be an error.

Thank you for pointing this out. We have corrected the value in Supplementary Table 2 from 7.3±0 to 7.3±1.2.

Reviewer #3: This review concentrates primarily on the statistical analysis. In general, the analysis seems sound. However, the statistical methods section should be rewritten slightly to clarify the various components of the statistical analysis.

Was the AUC used as a way of time-averaging the endpoint(s)? Please motivate the use of the AUC in this setting.

The Area Under the Curve (AUC) analysis was employed in this study as a method to comprehensively capture the temporal changes in outcomes throughout the entire study period. Rather than simply analyzing isolated timepoints, the AUC approach cumulatively reflected the total amount of effectiveness outcomes from randomization to the final follow-up. This methodology provides a more holistic evaluation of treatment effects over time by considering the entire trajectory of patient response.

More generally, note that each study assessment gives rise to one or more endpoints, which are used to assess the hypotheses of the study. With this in mind, please indicate the statistical methods section which endpoints are assessed via which statistical methods, as well as any transformations that are applied before statistical analysis (e.g., the AUC). This should help clarify the flow from data to results to conclusions.

Following your advice, we changed the statistical analysis section as follows:

During the study period, AUC analysis and survival analysis were conducted to comprehensively capture the temporal changes in outcomes. The AUC cumulatively reflected the total amount of effectiveness outcomes from randomization to the final follow-up and was calculated on a 1-week basis according to the trapezoidal rule. The AUC of the two groups were compared using Student's t-test. Furthermore, the proportion of patients (%) in each group was compared at each timepoint where VAS scores decreased to less than half of the baseline values. Kaplan–Meier curves were employed for survival analysis to compare the time until TMD pain ‘recovery’ was achieved, defined as pain outcomes decreasing to less than half of the baseline levels post-randomization, and the curves were analyzed using the log-rank test. Hazard ratios between the two groups were compared using the Cox proportional hazards model. A significance level of 0.05 was applied for this study, and all analyses were performed using SAS 9.4 (© SAS Institute, Inc., Cary, NC, USA) and R Studio 1.1.463 (© 2009–2018 RStudio, Inc.).

To further elaborate, generally, the structure should be rather parallel, with:

Hypotheses -> Assessments -> Endpoints -> Statistical methods -> Results -> Conclusions

Thank you for your valuable suggestion. To ensure clarity and coherence in the manuscript, we have explicitly incorporated the study hypothesis within the Study Design subsection.

Study design

This study was designed as a two-arm, multicenter, assessor-blinded RCT to evaluate the effectiveness and cost-effectiveness of CMT compared to usual care in patients with myofascial TMD. We hypothesized that CMT would result in greater pain reduction, improved jaw function, enhanced health-related quality of life, and favorable cost-effectiveness profile. Additionally, we hypothesized that CMT would lead to faster symptom relief and recovery compared to usual care.

This study was conducted from September 24, 2018, to June 29, 2019, at one university hospital (Kyung Hee University Korean Medicine Hospital, Gangdong) and four spine specialty Korean medicine hospitals (Jaseng Hospital of Korean Medicine in Gangnam, Daejeon, Bucheon, and Haeundae). The protocol of this RCT has been published [14] and was approved by the Institutional Review Board of the respective hospitals (JASENG 2018-06-008, 2018-06-010, 2018-06-011, and 2018-06-012, and KHNMCOH 2018-05-007). Written informed consent was obtained from all participants. This study was conducted according to the Consolidated Standards of Reporting Trials. The study timetable is presented in Table 1.

It would be useful to have figures that display longitudinal data --- these could be included just as supplementary figures.

Thank you for your helpful suggestion. We have added S2 Figure to display the longitudinal data as recommended. Additionally, we have revised the main text to reflect this addition, stating:

Longitudinal changes in outcomes, including pain(VAS), functional measures (JFLS-Global), and quality of life indicators (SF-12 PCS, EQ-VAS, EQ-5D), are visualized in S2 Figure.

7. PLOS authors have the option to publish the peer review history of their article (what does this mean?). If published, this will include your full peer review and any attached files.

Do you want your identity to be public for this peer review? For information about this choice, including consent withdrawal, please see our Privacy Policy.

Reviewer #1: No

Reviewer #3: No

---

## [Decision Letter · Decision Letter 2]

21 Mar 2025

Effectiveness and cost-effectiveness of Chuna manual therapy for temporomandibular disorder: A randomized clinical trial

PONE-D-24-26791R2

Dear Dr. Lee,

We’re pleased to inform you that your manuscript has been judged scientifically suitable for publication and will be formally accepted for publication once it meets all outstanding technical requirements.

Kind regards,

Essam Al-Moraissi

Academic Editor

PLOS ONE

Additional Editor Comments (optional):

Reviewers' comments:

Reviewer's Responses to Questions

**Comments to the Author**

1. If the authors have adequately addressed your comments raised in a previous round of review and you feel that this manuscript is now acceptable for publication, you may indicate that here to bypass the “Comments to the Author” section, enter your conflict of interest statement in the “Confidential to Editor” section, and submit your "Accept" recommendation.

Reviewer #1: All comments have been addressed

Reviewer #3: All comments have been addressed

2. Is the manuscript technically sound, and do the data support the conclusions?

Reviewer #1: Yes

Reviewer #3: (No Response)

3. Has the statistical analysis been performed appropriately and rigorously?

Reviewer #1: Yes

Reviewer #3: (No Response)

4. Have the authors made all data underlying the findings in their manuscript fully available?

Reviewer #1: Yes

Reviewer #3: (No Response)

5. Is the manuscript presented in an intelligible fashion and written in standard English?

Reviewer #1: Yes

Reviewer #3: (No Response)

6. Review Comments to the Author

Reviewer #1: Previously presented comments have been adequately modified.

The methods, results, and discussions described in this article are suitable for publication in this journal.

Reviewer #3: (No Response)

7. PLOS authors have the option to publish the peer review history of their article (what does this mean? ). If published, this will include your full peer review and any attached files.

**Do you want your identity to be public for this peer review?** For information about this choice, including consent withdrawal, please see our Privacy Policy .

Reviewer #1: No

Reviewer #3: No

---

## [Editor Report · Acceptance letter]

PONE-D-24-26791R2

PLOS ONE

Dear Dr. Lee,

I'm pleased to inform you that your manuscript has been deemed suitable for publication in PLOS ONE. Congratulations! Your manuscript is now being handed over to our production team.

Kind regards,

on behalf of

Dr. Essam Al-Moraissi

Academic Editor

PLOS ONE